# Nickel Oxide-Carbon Soot-Cellulose Acetate Nanocomposite for the Detection of Mesitylene Vapour: Investigating the Sensing Mechanism Using an LCR Meter Coupled to an FTIR Spectrometer

**DOI:** 10.3390/nano12050727

**Published:** 2022-02-22

**Authors:** Lesego Malepe, Patrick Ndungu, Derek Tantoh Ndinteh, Messai Adenew Mamo

**Affiliations:** 1Energy, Sensors and Multifunctional Nanomaterials Research Group, Department of Chemical Sciences, University of Johannesburg, P.O. Box 17011, Johannesburg 2028, South Africa; malepelesegomiccah@gmail.com (L.M.); pndungu@uj.ac.za (P.N.); dndinteh@uj.ac.za (D.T.N.); 2DST-NRF Centre of Excellence in Strong Materials (CoE-SM), University of the Witwatersrand, Johannesburg 2001, South Africa

**Keywords:** mesitylene, carbon nanoparticles, nickel oxide nanoparticles, nanocomposite, in situ FTIR, gas sensors

## Abstract

Nanocomposite sensors were prepared using carbon soot (CNPs), nickel oxide nanoparticles (NiO-NPs), and cellulose acetate (CA), which was used to detect and study the sensing mechanism of mesitylene vapour at room temperature. Synthesised materials were characterised using high-resolution transmission electron microscopy (HR-TEM), powder x-ray diffraction (PXRD), Raman spectroscopy, and nitrogen sorption at 77 K. Various sensors were prepared using individual nanomaterials (NiO-NPs, CNPs, and CA), binary combinations of the nanomaterials (CNPs-NiO, CNPs-CA, and NiO-CA), and ternary composites (NiO-CNPs-CA). Among all of the prepared and tested sensors, the ternary nanocomposites (NiO-CNPs-CA) were found to be the most sensitive for the detection of mesitylene, with acceptable response recovery times. Fourier-transform infrared (FTIR) spectroscopy coupled with an LCR meter revealed that the mesitylene decomposes into carbon dioxide.

## 1. Introduction

There is a high demand for environmental monitoring sensing devices for volatile organic compounds (VOCs), such as methane [1], methanol [2], ethanol [3], triethylamine [4], acetone [5], toluene [6], xylene, ethylbenzene [7], and the under-reported toxic mesitylene. The detection of mesitylene vapour is very important due to the fact that the inhalation of this vapour can result in various health concerns. Mesitylene is an organic compound, also known as 1,3,5-trimethylbenzene, with a vapour pressure of about 2.49 mmHg at room temperature. It is commonly found in mixed liquids of detergents, inks, and some paints. Thus, people working in an open or closed environment that involves painting cars or petroleum companies, as examples, are likely to be affected. People who worked in the painting industries were noted to have anemia and bronchitis as the paints contain about 30–50% 1,3.5-trimethylbenzene [8].

Both p-type and n-type semiconductor metal oxides (SMOs), including SnO_2_ [9], TiO_2_ [10], WO_3_ [11], ZnO [12], NiO [13], and In_2_O_3_ [14], are reported to be good sensing materials, and may be used in gas sensors due to their sensitivity and fast response-recovery times, low signal to noise ratio, and their good reproducibility. SMOs offer several advantages as gas sensors, such as their cost-effectiveness, ease of synthesis, and tunable and flexible morphology. Unfortunately, the biggest disadvantage with SMO as gas sensing components is that they usually work at relatively high temperatures, between 240 and 400 °C, and suffer selectivity issues towards the gas of interest. The high-temperature needed can have a negative impact on the growth of SMOs gas sensors as they result in short-life spans and high-power usage [9,10,11,12,13,14]. Nickel oxide (NiO) is reported to be one of the promising materials to be used in gas sensing based on its good sensing performances and good thermal stability. NiO is a p-type SMO with a bandgap of 3.6–4 eV, and it is also known to have catalytic properties [13,15,16].

Many methods have been studied and explored to further reduce the working temperature of the SM-based sensor from high temperature to room temperature. Methods that have been used include the introduction of carbonaceous materials (such as carbon nanotubes [17] and graphene [18]) and the use of conducting polymers (such as polyaniline [19], or polypyrrole [20]). Interestingly, a less reported material is carbon soot prepared from the pyrolysis of ordinary household wax candles. Carbon soot represents a material that can be investigated as an integral component of gas sensors due to its high conductivity, inexpensive nature, low toxicity, good porosity, and good stability. Carbon soot is known to be composed of carbon nanoparticles (CNPs) based on their spherical appearance or onion ring like structure [21]. Conducting polymers are mostly explored in gas sensors to improve sensitivity, selectivity, and to reduce the high working temperature of SMOs. However, cellulose acetate (CA) as a non-conducting polymer improves sensing performance sensitivity, selectivity, and response-recovery time [22]. In addition, the excellent characteristics from CA include nontoxicity, high surface area, high absorption capacity, inexpensive nature, and biodegradability [22]. Although CA has good attributes to be used in gas sensing, it is rarely reported. In this work, NiO nanoparticles, CNPs, and CA composite are used to detect mesitylene at room temperature. Furthermore, the LRC meter was coupled with a Fourier transform infrared spectroscopy (FTIR) instrument to investigate the sensing mechanism of mesitylene.

## 2. Materials and Methods

### 2.1. Material and Reagents

Lighthouse white Candles (purchased from a local supermarket) were used for the preparation of carbon nanoparticles. Cellulose acetate, sodium hydroxide, nickel (II) hexahydrate, mesitylene, and N-N-dimethylformamide (99.8%) were purchased from Sigma-Aldrich (Kempton Park, South Africa). Ethanol (99.8%) was purchased from Merck (Modderfontein, South Africa).

### 2.2. Preparation of Carbon Nanoparticles

The carbon nanoparticles (CNPs) soot was synthesized following methods reported by Olifant et al. [22]. In brief, the CNPs were prepared by placing a ceramic cup above a burning candle (~1 cm above the candle) to collect the black smoke (soot). After 30 min, the black product was scraped from the ceramic cup and stored in a glass vial until ready for further use.

### 2.3. Synthesis of Nickel Oxide Nanoparticles (NiO NPs)

Nickel oxide nanoparticles were synthesised following the synthetic method reported by Rahdar et al. [23]. Firstly, 5.94 g nickel chloride (II) hexahydrate was dissolved in 250 mL deionized H_2_O at room temperature. The solution was then magnetically stirred at 50 °C for 40 min to obtain a homogeneous solution. A pre-prepared 10 molar solution of NaOH was added dropwise to the homogeneous nickel chloride (II) hexahydrate solution to maintain pH 8, then the solution was stirred for a further 2 h at 50 °C. Thereafter, green precipitates were formed, recovered, and then washed several times interchangeably with ethanol and deionized H_2_O. The product was then dried at 60 °C overnight, and finally annealed at 500 °C for 2 h to obtain NiO NPs.

### 2.4. Preparation of NiO-CNPs-CA Composite

The NiO-NPs were combined with the CNPs and cellulose acetate (CA) to make a composite (NiO-CNPs-CA). The composite was prepared by mixing 1.00 g NiO NPs, 1.00 g CNPs, and 3.00 g CA polymer in 100 mL N,N-dimethylformamide. The mixture was then sonicated for 10 min and then physically mixed using a magnetic stirrer for the formation of the NiO-CNPs-CA composite under stirring for 48 h at room temperature. 

### 2.5. Gold Plated-Interdigitated Electrode and Sensor

The gold plated-interdigitated electrode was designed to have 0.1 mm width spacing between gold lines of 18 pairs of 7.9 mm long as shown in Figure 1a. The electrode was cleaned through washing with EtOH followed by DMF, then allowed to dry at room temperature. Figure 1b shows the appearance of a prepared sensor.

### 2.6. Characterization

The morphology of the prepared nanoparticles was characterized using high-resolution transmission electron microscopy at an acceleration voltage of 200kV, JEOL-TEM 2010 (JEOL, Tokyo, Japan) using Gatan software (3 View 2XP, Gatan Inc, Tokyo, Japan). Samples were dispersed onto copper grids for the TEM analysis. Scanning electron microscopy was performed at 30kV with a FEI Nova Nanolab 600 (FEI, Hillsboro, OR, USA). Structural analysis was performed using powder X-ray diffraction (PXRD) on a Bruker D2 Phaser using LynxEye detector (Bruker, Billerica, MA, USA) with radiation of a CuKα at a wavelength of 0.154 nm. A Bruker Senterra laser Raman spectrometer (Bruker, Karlsruhe, Germany) fitted with frequency-doubled Nd-YAG laser with the wavelength of 532 nm, was used for Raman analysis. Nitrogen-sorption experiments were performed on a Micrometrics ASAP 2020 (Micrometrics GmbH, Unterschleissheim, Germany) at 77 k to determine the textural characteristics of the materials. For the nitrogen sorption experiments, before each analysis, 0.100 g of the prepared samples were degassed using N_2_ gas for 12 h at 100 °C. Fourier transform infrared spectroscopy (FTIR, Bruker-Alpha, Leipzig, Germany) was used to study the sensing mechanism. X-ray photoelectron spectroscopy (XPS, XSAM800, Kratos, Manchester, UK) was used to determine the oxidation state and the elemental analysis. 

### 2.7. Gas Sensing Setup and Sensors Testing

The electrical characterization of the sensor was done using a GW Instek 8101G LRC meter connected to our fabricated sensor and the computer (see Figure 2). A 0.5 AC input signal was supplied to the sensor at a relative frequency of 25 kHz at room temperature [21]. The sensor was placed in a 20 L volumetric flask connected to a vacuum pump to remove any internal gases within the system. Tested sensing materials used to fabricate sensors for the detection of mesitylene included CNPs, CA, NiO nanoparticles, and their relative mixtures. The mass ratio of the CNPs to CA polymer was kept 10:30 mg all-time. However, the mass ratio of the NiO nanoparticles within the ternary composite was changed from 10, 30, and 60 mg, respectively, so as to investigate the effect of metal oxide within the composite as a function of sensor optimization. The mixture of the desired materials was dispersed in 10 mL of N,N-dimethylformamide (DMF) and then 10 μL was drop-coated onto an interdigitated gold electrode. Prepared sensors were allowed to dry at room temperature under a vacuum to remove any excess DMF. The prepared sensors were left in a desiccator for 5 weeks or more before use. Prepared sensors were labelled as Snr 1: NiO, Snr 2: CA, Snr 3; CNPs, Snr 4: NiO-CA (10:30 mg mass ratio), Snr 5: CNPs-CA (10:30 mg mass ratio), Snr 6: NiO-CNPs (10:10 mg mass ratio), Snr 7: NiO-CNPs-CA (10:10:30 mg mass ratio), Snr 8: NiO-CNPs-CA (30:10:30 mg mass ratio) and NiO-CNPs-CA (60:10:30 mg mass ratio). Five tests were performed on one sensor, wherein 1, 2, 3, 4, and 5 μL volumes of mesitylene were injected into a 20 mL volumetric flask at a time. Contact time between the mesitylene vapour and the sensors was 16 min, and the gas was removed using a vacuum pump in the presence of atmospheric air for 3 min. The sensor was allowed to recover for 3 min before the next injection. Sensors were tested at room temperature with a relative humidity of about 42%.

The concentration of the injected analyte was calculated using the following formula:
$$C=\frac{22.4\;p{\mathrm{TV}}_{s}}{273\;M_{r}V}\times 1000,$$
where *C* is the concentration of mesitylene at room temperature, V_s_ is the volume injected, V is the volume size of the volumetric chamber, M_r_ and *p* represent the molar mass and density of the mesitylene respectively [22]. 

### 2.8. In Situ FTIR-LRC Meter Testing Setup

The LRC meter was connected to the sensor through electrical wires and placed inside the cylindrical gas cell. A 110 mL cylindrical gas cell with two KBr windows was placed inside the FTIR (PerkinElmer Spectrum 100, Waltham, MA, USA) sample holder. Mesitylene vapour was exposed to the sensor that was placed in the cylindrical gas cell and the FTIR scan was taken every two minutes after exposure as shown in Figure 3 and electrical measurements during the interaction between the sensor and the analyte were taken. 

### 2.9. Curve Fitting for the Infrared CO_2_ Band

The area under the curve was obtained from fitting the IR bands using magic plot software and a Gaussian function. Six spectra were used from the first minute to the eleventh minute after mesitylene exposure. The data chosen were from 672 to 661 cm^−1^. For all areas under the curve for all spectra, the area at *t* = 0 min was subtracted due to a fact that it might be the atmospheric CO_2_. The equation used is:Δ*Ac* = *AC*_*t*=*i*_ − *AC*_*t*=0_
where Δ*A**c* is the relative area under the IR CO_2_ curve, *AC*_*t*=*i*_ is the area under the curve containing the atmospheric CO_2_ and *AC*_*t*=0_ the area under the curve *t* = 0 min.

## 3. Results and Discussions

### 3.1. Microscopy

High-resolution transmission electron microscopy (HRTEM) was used to reveal morphological details of the CNPs, NiO NPs, and NiO-CNPs-CA composite, as shown in Figure 4. HRTEM images shown in Figure 4a,b, revealed that the CNPs are generally spherical in structure, and include separate spherical nano-sized carbon particles, fused spherical structures, and some nanoparticles were stacked on top of other structures. The carbon soot nano-spheres are stacked on each other and some are fused, forming chain-linked spheres that resulted from the deposition of soot, as clearly shown in Figure 4b. The diameter of the synthesised CNPs was between 10 and 50 nm, and the average particle size was 30 nm (Figure 4e). Similar findings were reported by Olifant et al. in their studies [21]. The NiO nanoparticles morphology was characterized using HR-TEM as shown in Figure 4c. It was found that the NiO were mainly cubic particles but with some spherical and hexagonal structures, and the cubic shapes observed correlates well with NiO nanoparticles powder X-ray diffraction spectroscopy (PXRD) shown in (see in the PXRD section). The diameter of the NiO nanoparticles ranged between 20 and 90 nm, as shown on the particle size distribution graph in Figure 4f. Particles with diameters of about 55 nm are dominant within the NiO prepared sample.

The SEM images revealed that the CNPs are highly agglomerated, as expected, and form some lumps as shown in Figure 5a. The EDS in Figure 5b revealed that the CNPs are composed of about 91% carbon and 9% oxygen.

### 3.2. X-ray Photoelectron Spectroscopy

The surface oxygen content on the nanomaterials plays a crucial role in gas sensing [24]. Thus, XPS analysis was used to reveal the availability of oxygens present on materials, the oxidation state of nickel, and the chemical composition of NiO-CNPs-CA composite as presented in Figure 6. The XPS survey spectrum of NiO-CNPs-CA reveals the existence of Ni, C, and O (see Figure 6a). The O 1 s XPS spectra indicate that both NiO and NiO-CNPs-CA have all oxygen species occurring at 529.3 eV, 532 eV and 533 eV assigned for O_α_, O_β,_ and O_γ_ respectively. However, CNPs present only O_β_ and O_γ_ (see Figure 6b–d), the O_α_ represents the lattice oxygen species, O_β_ is designated for surface adsorbed oxygen species, and O_γ_ represents the availability of adsorbed OH on the materials [25]. Figure 6e,f present Ni 2p XPS spectra, revealing the existence of peaks at binding energies 855.5 eV and 873.4 eV representing Ni 2p_3/2_ and Ni 2p_1/2_, respectively, and the separating energy is 17.5 eV. Furthermore, both the Ni 2P spectra have satellite peaks occurring at approximately 861 eV (Ni 2p_3/2_) and 878.8 eV (Ni 2p_1/2_). The positioning of the main and satellite peaks indicates the presence of Ni^2+^ in both NiO and NiO-CNPs-CA composite [24,25,26,27].

### 3.3. Powder X-ray Diffraction

Figure 7 shows the XRD analysis of carbon nanoparticles, cellulose acetate, NiO nanoparticles, and NiO-CNPs-CA composite. Figure 7a shows the XRD pattern of carbon nanoparticles exhibiting two broad diffraction peaks at appearing at 2*θ* = 25.2° and 44° ascribed to the crystal planes of (002) and (101), which are assigned to the graphitic and amorphous nature of the CNPs (ICDD: 04-018-7559).Figure 7b shows the X-ray diffraction peaks of cellulose acetate occurring at 2*θ* = 9.5° and 17.3°. Figure 7c shows the XRD pattern of the nickel oxide nanoparticles, with diffraction peaks positioned at 2*θ* = 37.3°,43.3°, 62.9°, 75.4°, and 79.4°. These peaks are indexed to the crystal planes of (111), (200), (220), (311), and (222) reflections of the cubic phase of NiO (JCPDS, ICDD no. 78–0423). The average NiO nanoparticles crystallite size was calculated using D = 0.9λ/βCos*θ* (Debye-Scherrer formula), where D represents the average crystallite size, λ is the radiation wavelength of CuKα (0.15406 nm), β is the full width recorded at half maximum intensity (FWHM), and *θ* represents the diffraction angle. The average NiO nanoparticles crystallite size was found to be 46 nm using crystal panes (111), (200), or (222). The formation of the NiO-CNPs-CA composite was characterized using XRD as shown in Figure 7a There is a presence of low-intensity diffraction peaks occurring at 2*θ* = 9.5° and 17.3° which are assigned to the cellulose acetate within the composite. There is an absence of clear diffractions peaks from the carbon nanoparticles. However, there is a broad low intensity peak (~18–25° 2*θ*) within the composite spectra which can be ascribed to the CNPs. The excellent crystallinity of NiO NPs within the composite shows sharp high intensity peaks, as expected.

### 3.4. Raman Spectroscopy

Figure 8 displays the Raman spectra of cellulose acetate, carbon nanoparticles, nickel oxide, and NiO-CNPs-CA composite. Raman spectra for cellulose acetate showed peaks at 1736, 1435, and 1382 cm^−1^ attributed to a carbonyl group (C=O) and carbon to hydrogen (C–H) stretching [28] from the acetyl groups (Figure 8a). The Raman shifts exhibited at 660, 830, 902, and 978 cm^−1^ are attributed to C–O–H, O–H, C–H, and C–O stretching for CA [28]. Carbon nanoparticles prepared from pyrolysis of the candle show two broad Raman shifts at about 1350 and 1555 cm^−1^. These peaks are assigned to the D and G bands respectively as presented in Figure 8b. The D band is indicative of defect structures, usually from amorphous carbons present within the sample. Meanwhile, the G band shows the graphitic nature of samples [29]. The Raman spectrum of synthesized nickel oxide nanoparticles (NiO NPs) are presented in Figure 7c. The Raman shift at 580 cm^−1^ (from LO mode) and 1090 cm^−1^ are assigned for one phonon (1P) and the signal positioned 980 cm^−1^ (from TO + LO mode) is assigned for two magnons (2M) at excitations which signify the 3-dimensional nature of the NiO (cubic) and antiferromagnetic property. One phonon peak at 580 cm^−1^ may correspond to the single-crystal structure of the nickel oxide nanoparticles [30]. Figure 7e presents the Raman vibrations of NiO-CNPs-CA composite. The presence of the D and G bands confirmed the presence of carbon nanoparticles in the composites. The labelled area from 700 to 1400 cm^−1^ shows the presence of cellulose acetate.

### 3.5. Textural Characteristics

The nitrogen adsorption-desorption isotherms were used to determine the BET (Brunauer-Emmett-Teller) surface area, pore size distribution, and average pore volume. Figure 9 shows nitrogen adsorption-desorption of CNPs, CA, NiO NPs, and NiO-CNPs-CA. The CNPs have a higher BET surface area of 125.28 m^2^/g and a pore volume of 0.53 cm^3^/g, while cellulose acetate has a lower BET surface area of 2.38 m^2^/g, a pore volume of 0.01 cm^3^/g, and the NiO NPS had a BET surface area of 12.09 m^2^/g and a pore volume of 0.13 cm^3^/g. The NiO-CNPs-CA had a BET surface area of 0.74 m^2^/g and a pore volume of 0.0027 cm^3^/g. The NiO-CNPs-CA material showed a decrease in BET surface area and pore size as compared to single sensing materials, which suggests that metal oxides and the cellulose polymer fill the pores that were formed between the carbon nanoparticles [21,22]. A decrease in BET surface area and pore size plays a crucial role in the sensitivity of the analytes during the sensing process [31].

### 3.6. Performances of the Sensors for Mesitylene Sensing

The electrical performance of the fabricated sensors was investigated for the prepared sensors and measurements were taken at room temperature and 42% RH. The three sensing materials, NiO, CNPs, and CA, were used to fabricate all the sensors. A total of 9 sensors were fabricated. Three of the sensors were made with a single sensing material, sensors (Snr) 1, 2, and 3 were NiO, CA, and CNPs respectively. Sensors with two sensing materials, were fabricated, and were labelled (Snr) 4, 5, and 6. The sensors labelled Snr 4, 5, and 6 were prepared with NiO-CA with a mass ratio of 10:30 mg, CNPs-CA with a mass ratio of 10:30 mg, and NiO-CNPs with a mass ratio of 10:10 mg. Finally, three sensors were fabricated composed of all three sensing materials with a variable mass of NiO; Snr 7, 8, and 9, were NiO-CNP-CA with the mass ratio (NiO: CNP: CA) 10:10:30 mg, 30:10:30 mg, and 60:10:30 mg respectively.

The relative resistance (ΔR) measurement was recorded for all the fabricated sensors while the sensors were with (gas in) and without (gas out) exposure of the analyte. The concentration of the analyte was increased from 8.8 to 43.9 ppm. Sensors based on single sensing material, NiO, CA, and NiO-CA, did not show any meaningful response towards the analyte since the signal-to-noise ratios were very high. The sensors with two sensing materials, only the sensors with CNPs-CA and NiO-CNPs (Snr 5 and 6), displayed better responses than the single sensing material. The sensor’s response did show some incremental while the sensors were in contact with the analyte vapour, and when the analyte vapour was removed from the system the response of the sensor returned to the baseline signal, which is an indication of the sensor’s ability to regenerate by themselves. Unlike the three sensing materials-based sensors, the two sensing materials-based sensors did not show any linearity as the concentration of the analyte increased (see Figure 10). However, sensors based on three sensing materials performed well with predictable linearity; specifically, as the concentration increased the relative response also increased (see Figure 11). In addition, the sensors (Snr) 7, 8, and 9 displayed response–recovery curves (similar to the two sensing material based sensors) as the sensors were exposed to the analyte vapour and recovered to the baseline when the analyte vapour was removed from the system. Response time is a time at which a sensor takes from its baseline to the maximum response (saturation) at a specific amount of gas analyte. However, in this study, we consider the response times at which 90% of the maximum response and recovery time for all the analytes was recorded at 90% recovery of the sensors from its maximum response to the baseline and basically, recovery time is the time a sensor takes for a corresponding gas maximum response to return to its baseline [32]. Although the three sensors performed well, the performance of each sensor was not the same. The Snr 9 sensor had excellent performance with a sensitivity of 0.27 Ω ppm^−1^, a response time of 125 s, and a recovery time of 72 s. This was followed by Snr 7 with sensitivity at 0.24 Ω ppm^−1^, a response time of 72 s, and a recovery time of 19 s (see Figure 12a and Table 1) which is faster than Snr 9. Snr 8, however, was the poorest sensor in terms of sensitivity towards the analyte vapour, although the response and recovery times were the fastest compared to any of the other sensors.

### 3.7. In Situ FTIR-LCR Meter Study

From the response curve, all of the sensors exhibit p-type behaviour typical of semiconductor metal oxides interacting with reducing gases. The generally accepted sensing mechanism for semiconductor metal oxides is the process is usually initiated by oxygen molecules that are adsorbed on the surface of the sensing materials. Those oxygen molecules trap and react with electrons from the sensing materials conduction band and form a depletion layer containing negative oxygen species (O^2−^, O^−^, O_2_^−^) around the surface of the sensing materials. When the sensing materials are exposed to a reductive organic molecule, it reacts with reactive oxygen species on the surface of the sensing materials to form CO_2_ and H_2_O molecules, and the electrons flow back to and recombine with a certain number of holes, which causes a rapid increase in the resistance [33,34,35]. According to our results, when the individual sensing materials were exposed to the analyte vapour, they did not respond at all, likely due to the low temperature used (ambient temperature). However, only the sensors based on either the three or two (CNPs-CA and NiO-CNPs) sensing materials did respond when exposed to the analyte vapour. This result could be due to the presence of molecular oxygen on the surface of the carbonaceous materials that was adsorbed and converted into different reactive oxygen species, which has been previously reported to occur on various carbon blacks at 300 K [36]. The EDS results showed that the CNPs had a significant amount of oxygen, about 9% (Figure 5b), and the Raman analysis confirmed the CNPs contained amorphous carbons, which is similar to earlier reports on candle soot derived CNPs possessing carbon-oxygen surface species [37]. This finding is further confirmed by XPS analysis (see Figure 6b) the presence of the reactive oxygen species on the CNPs surface, similar to those reactive oxygen species on metal oxides, is the main reason the sensing mechanism goes through a deep oxidation mechanism. After making the composite, those reactive oxygen species still existed on the surface of the composite (see Figure 6d). The existence of oxygen species on the CNPs suggests that the CNPs do catalyse the detection of the mesitylene vapour on the CNPs-NiO and CNPs-CA composites (Figure 10), and oxygen species on the surface plays a critical role. Furthermore, our results clearly show that the existence of a synergistic effect between the NiO, CNPs and CA in the composite results in excellent responses (Figure 11). 

The in-situ FTIR coupled with LCR meter connected to Snr 7 (NiO-CNPs-CA) was chosen to study the sensing mechanism of the gaseous 1,3,5-trimethylbenzene. This was mainly due to Snr 7 better response and recovery time when compared to Snr 8 and 9, and it also had a relatively close relative response with Snr 9. During the in-situ FTIR coupled with LCR meter measurement, as the sensor was exposed to the analyte vapour, the FTIR spectra were collected every 2 min and, at the same time, the relative response was recorded. In this study, we focused on a specific CO_2_ IR band that was not saturated during the measurement and most importantly does not overlap with other IR bands, which can give us a clear understanding of the sensing mechanism. 

As the in-situ measurement progressed (see Figure 13a), a new distinct CO_2_ IR band at 668 cm^−1^ was detected and the intensity of the band grew as the exposure time increased. This IR band was assigned to the CO_2_ bending mode [38,39]. At the first interaction time (1 min), the intensity of the CO_2_ band was relatively weak. However, as the interaction time increases, the intensity of this particular band increased incrementally (Figure 13b). For easy visualization, we calculated the area under the curve of the CO_2_ IR band and plotted this as a function of time. The results clearly showed that the intensity of the CO_2_ IR band increased as the exposure time increased (see Figure 13d). Indeed, this relationship was further confirmed by looking at the relative resistance response of the sensor during the exposure. Before any interaction just before *t* = 0, the relative response of the sensor was low ΔR ≈ 0. However, as the analyte vapour started to interact with the sensor, the relative resistance response of the sensor increased suddenly (between 0 and 1 min), and then the progression slowed (Figure 13c). This increase in the relative resistance response can be directly related to the intensity of CO_2_ [40].

The combination of the three sensing materials in the composite results in the sensors performed better than a single sensing material and two sensing materials combined. However, results revealed that the sensors containing CNPs undergo total oxidation and convert the gaseous analyte into carbon dioxide, even if there is an absence of metal oxides [22]. The high content of atomic oxygen on the surface of CNPs, about 9% (see Figure 5b), was formed during the pyrolysis of light candles in the presence of atmospheric air. This result is in agreement with previously published work on carbon nanoparticles produced from candle soot [36]. The presence of atomic oxygen on the surface of CNPs did result in a highly disordered carbon structure, and this was confirmed with the Raman analysis of the I_D_/I_G_ ratio of 0.9. The high activity of charge transfers and the disordered nature of the CNPs promotes the chemisorption of the analyte vapour, and this increases the electrical response in chemical sensors [39]. Thus, the CNPs are key to the functionality of the sensors at ambient or room temperature.

Generally, it is known that the type of oxygen species on the surface determines whether the sensing mechanism will undergo total oxidation or oxidative dehydrogenation [41,42,43,44,45]. From our FTIR results, the mechanism undergoes deep oxidation due to the incremental formation of CO_2_. This might be due to the dominance of the electrophilic oxygen species on the surface of NiO and CNPs, and the synergistic interaction between the CNPs and NiO. Therefore, the possible chemical reaction between the oxygen species and mesitylene vapour (1)–(4) [46,47,48]:O_2_(gas) → O_2_(ads)(1)
O_2_(gas) + e− → O_2_−(ads)(2)
O_2_−(ads) + e− → 2O−(ads)(3)

Mesitylene:C_9_H_12_ (gas) + 24O− → 9CO_2_ + 6H_2_O + 24e−(4)

### 3.8. Humidity and Reproducibility Studies

Figure 14 shows the humidity studies of snr 9 for mesitylene vapour. Identical concentrations were used on these investigations. Specifically, 52.7 ppm of mesitylene vapour was used at these relative humidities (RH): 45%, 52% and 61%. The maximum response level at 45%, 52% and 61% RH was found to be 25, 25.5, and 26.3 Ω respectively. There is a slight increase in response as humidity increases, which could be due to water molecules affecting the response. 

Our sensors show good reproducibility (see Figure 15). Sensor 9 was used for reproducibility investigations wherein identical concentrations of mesitylene vapour were used, where 25.5 Ω was tested for 5 trials and the sensor showed almost equal maximum responses.

## 4. Conclusions

The successful synthesis of NiO NPs, CNPs, and NiO -CNPs-CA was achieved and the materials were characterized with HR-TEM, BET, P-XRD, and Raman spectroscopy. It was observed that single material NiO NPs, CA, and NiO-CA binary composite were not responsive towards mesitylene at room temperature, while CNPs, NiO-CNPs, CNPs-CA, and NiO-CNPs-CA respond. However, although the prepared sensors were different in terms of sensitivity towards the analyte vapour, it was observed that the 6:1:3 mass ratio of NiO-CNPs-CA sensor (Snr 9) had the highest sensitivity among all the prepared sensors, while the 1:1:3 mass ratio of NiO-CNPs-CA sensor (Snr 7) had the fastest response-recovery time. In-situ, the FTIR-LRC meter revealed the sensing mechanism of mesitylene vapour, whereby the reaction undergoes total decomposition to form CO_2_ and H_2_O.

## Figures and Tables

**Figure 1 nanomaterials-12-00727-f001:**
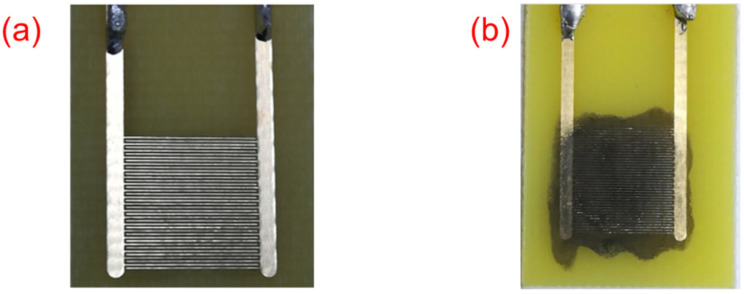
(**a**) gold-plated interdigitated electrode and (**b**) a prepared sensor.

**Figure 2 nanomaterials-12-00727-f002:**
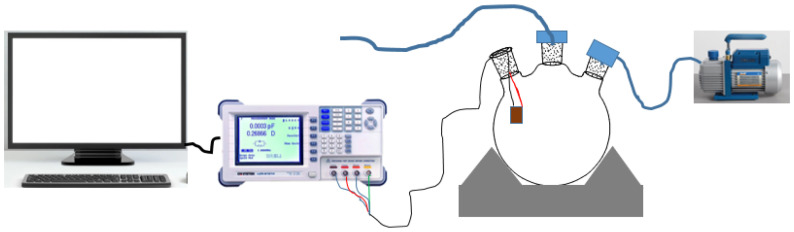
Illustration of the gas sensing setup used in this study.

**Figure 3 nanomaterials-12-00727-f003:**
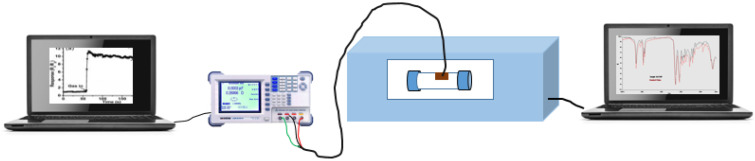
A diagram showing a combined online in situ FTIR and LCR meter setup for sensor testing.

**Figure 4 nanomaterials-12-00727-f004:**
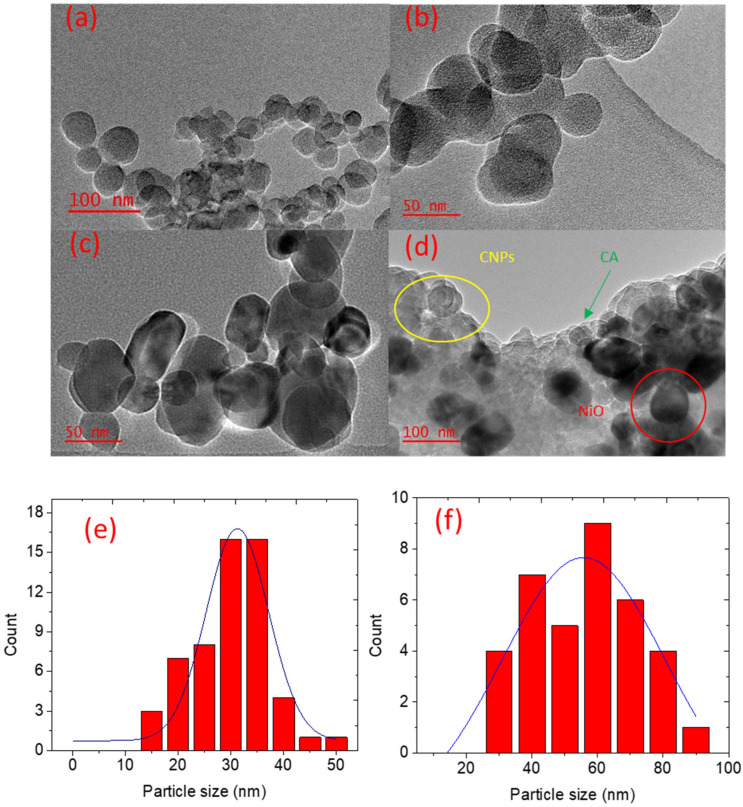
HR-TEM images of (**a**) CNPs and (**b**) CNPs at higher magnification, (**c**) NiO, (**d**) 1:1:3 mass ratio NiO-CNPs-CA, and the size distribution graph of (**e**) CNPs and (**f**) NiO.

**Figure 5 nanomaterials-12-00727-f005:**
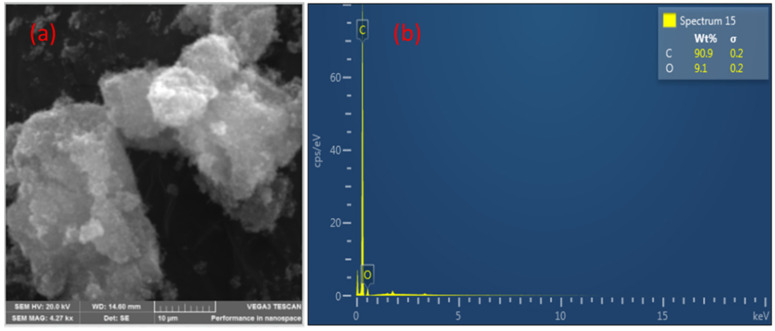
SEM image of (**a**) CNPs and EDS image of CNPs (**b**).

**Figure 6 nanomaterials-12-00727-f006:**
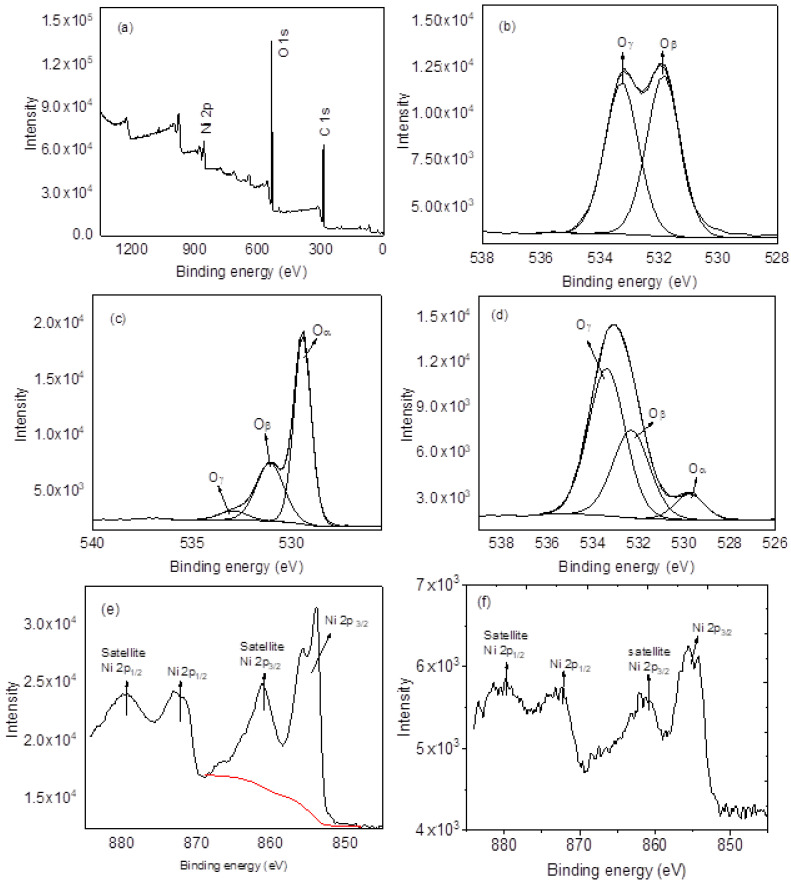
XPS spectra, (**a**) survey spectrum of NiO-CNPs-CA, (**b**) 1 O s of CNPs, (**c**) 1 O s of NiO, (**d**) 1 O s of NiO-CNPs-CA, (**e**) Ni 2p of NiO and (**f**) Ni 2p of NiO-CNPs-CA.

**Figure 7 nanomaterials-12-00727-f007:**
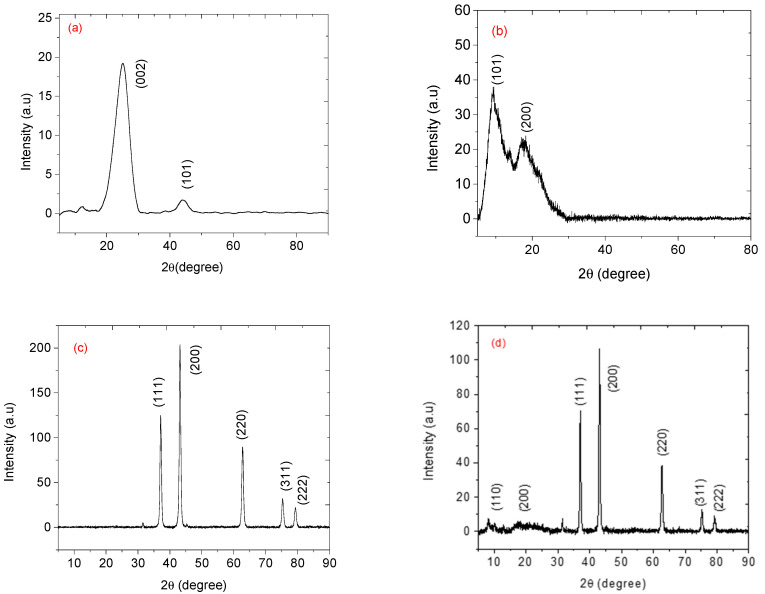
XRD patterns of (**a**) CNPs, (**b**) CA, (**c**) NiO NPs and (**d**) 1:1:3 mass ratio NiO NPs-CNPs-CA composite.

**Figure 8 nanomaterials-12-00727-f008:**
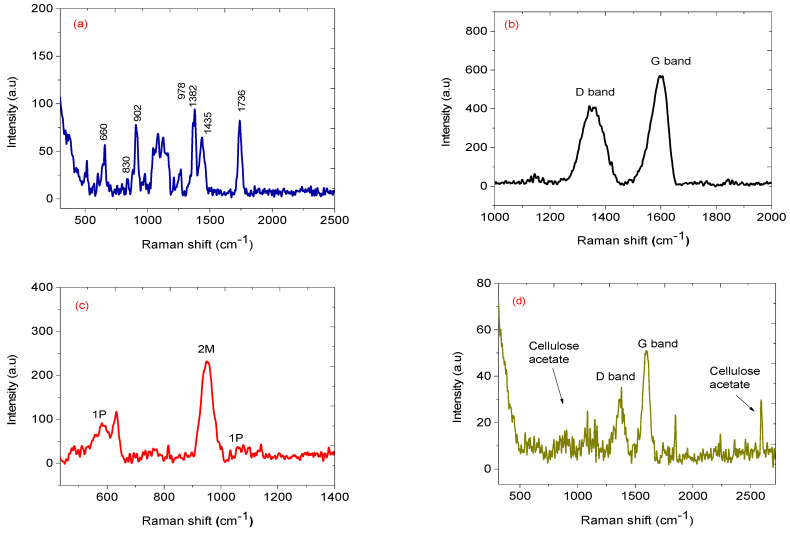
Raman spectra of (**a**) CA, (**b**) CNPs, (**c**) NiO and (**d**) 1:1:3 mass ratio NiO-CNPs-CA.

**Figure 9 nanomaterials-12-00727-f009:**
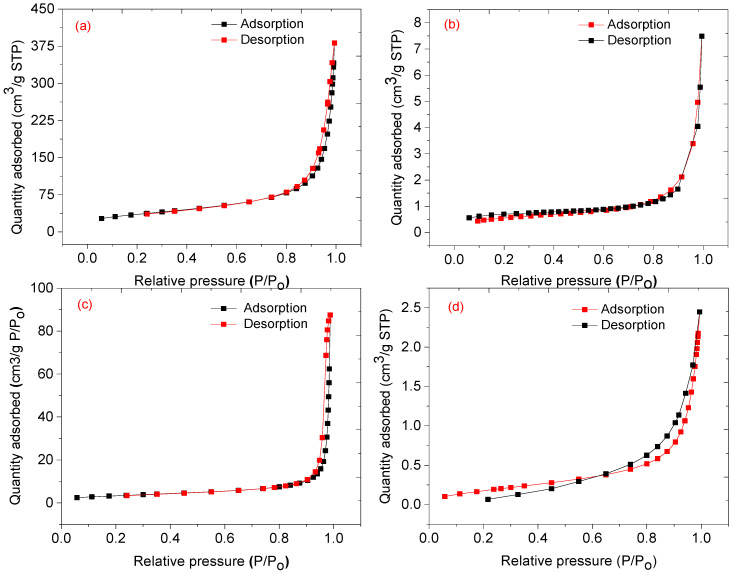
Nitrogen adsorption-desorption isotherms of (**a**) CNPs, (**b**) CA, (**c**) NiO nanoparticles, and (**d**) 1:1:3 mass ratio NiO-CNPs-CA.

**Figure 10 nanomaterials-12-00727-f010:**
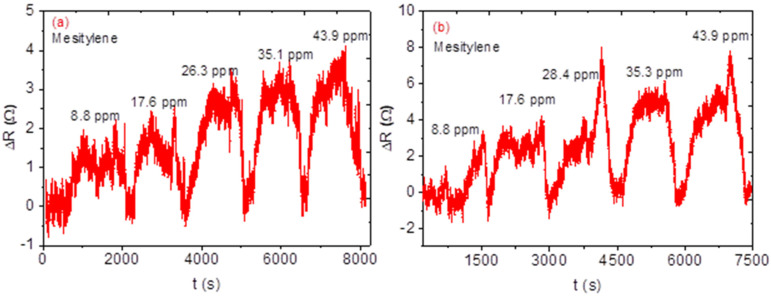
Dynamic response-recovery curve ΔR response as a function of time at variable concentrations of (**a**) CNPs-CA (**b**) NiO-CNPs.

**Figure 11 nanomaterials-12-00727-f011:**
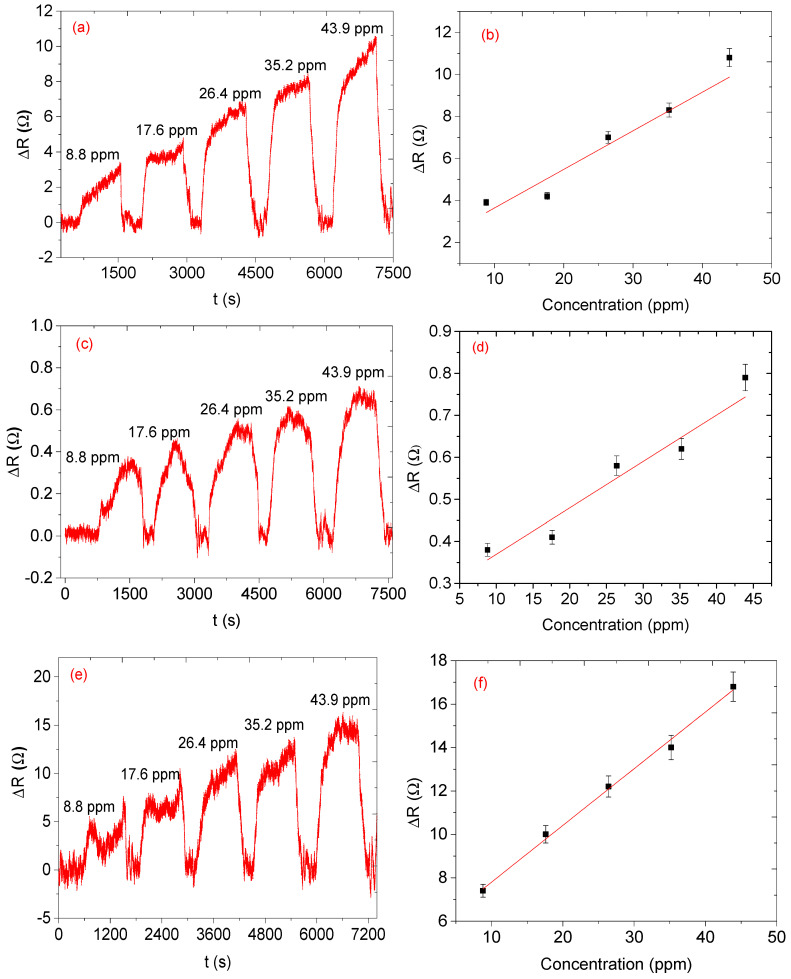
Dynamic response-recovery curve ΔR response as a function of time at variable concentrations of NiO-CNPS-CA sensor with a mass ratio of (**a**) and (**b**) 1:1:3 (Snr 7), (**c**) and (**d**) 3:1:3 (Snr 8) and (**e**) and (**f**) 6:1:3 (Snr 9).

**Figure 12 nanomaterials-12-00727-f012:**
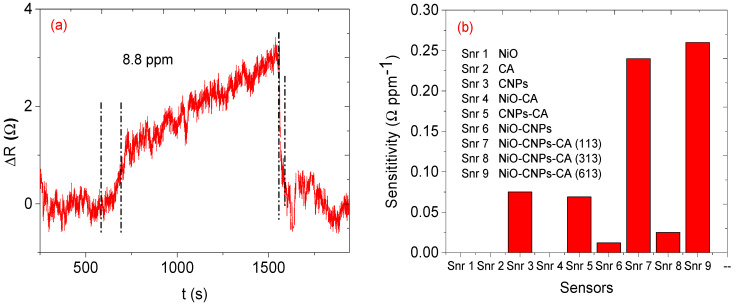
(**a**) Response-recovery time of Snr 7 and (**b**) sensitivity graph for mesitylene.

**Figure 13 nanomaterials-12-00727-f013:**
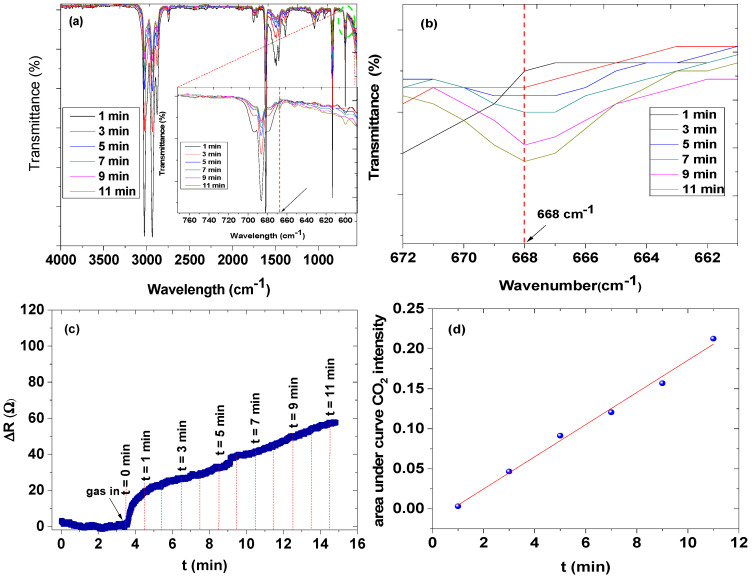
The in situ FTIR of mesitylene (**a**) the in situ FTIR CO_2_ bands in a narrow range (**b**), the mesitylene vapor response curve online with the FTIR (**c**), and the areas under the curve of the CO_2_ IR intensities against time (min) (**d**).

**Figure 14 nanomaterials-12-00727-f014:**
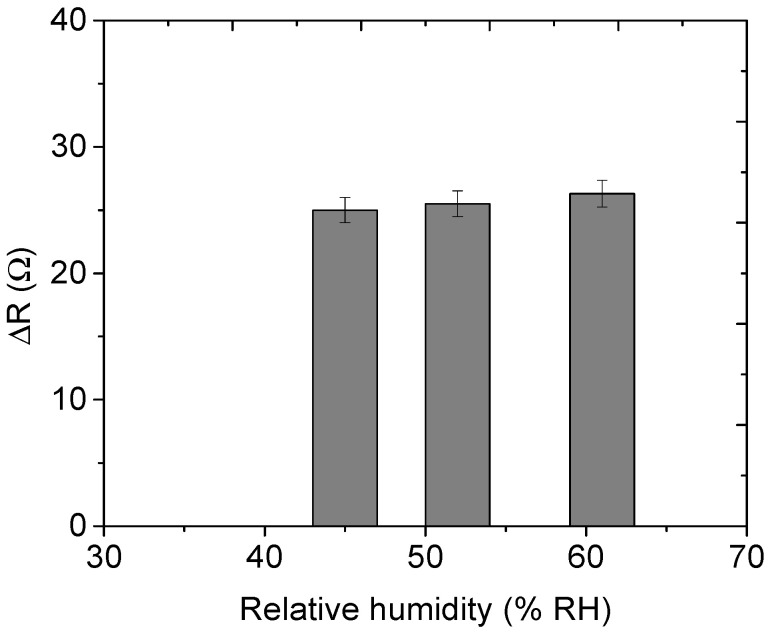
Humidity investigations on sensor 9.

**Figure 15 nanomaterials-12-00727-f015:**
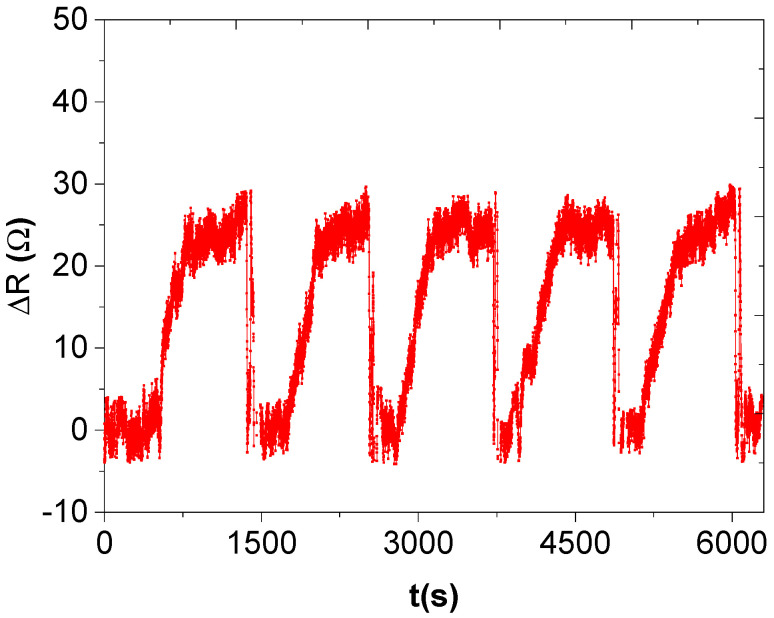
Repeatability graph of sensor 9 on mesitylene vapour.

**Table 1 nanomaterials-12-00727-t001:** Response and recovery time of the sensors during mesitylene detection.

Sensors	Response Time (s)	Recovery Time (s)
Snr 1: NiO	-	-
Snr 2: CA	-	-
Snr 3: CNPs	97	44
Snr 4: NiO-CA	-	-
Snr 5: CNPs-CA	86	39
Snr 6: NiO-CNPs	84	51
Snr 7: NiO-CNPs-CA (1:1:3)	72	19
Snr 8: NiO-CNPs-CA (3:13)	37	51
Snr 9: NiO-CNPs-CA (6:1:3)	125	72

## Data Availability

Not applicable.

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
