# Peer review of "Nickel Oxide-Carbon Soot-Cellulose Acetate Nanocomposite for the Detection of Mesitylene Vapour: Investigating the Sensing Mechanism Using an LCR Meter Coupled to an FTIR Spectrometer"

_nanomaterials, 2022, doi:10.3390/nano12050727_

Round 1
Reviewer 1 Report
The characterisation of the materials and the study of the detection mechanism is comprehensive and rigorous, but the characterisation of sensor performances has weaknesses and a lack of rigour.
Firstly, there are a few mistakes to be corrected:
Line 43 : “low signal to noise ratio” I think it's more about high.
Line 54: “In order to lower the high working temperature.“ This is not a sentence.
Line 258: “pore size plays a crucial role in the selectivity” I think it's more about sensitivity.
Secondly, several issues need to be addressed:
Lines 79-83: Are the physical properties (size distribution, aggregates size…) of the collected CNP reproducible? How did the authors ensure this? Possible influence of ambient humidity?
3.2. Gas sensing setup, sensor fabrication, and testing
Please add a picture of a sensor and specify its dimensions, the width and spacing of the electrodes, the thickness of the sensing layer.
What is the advantage of using an AC voltage at 25kHz and an RLC bridge to measure only the sensor resistance (as shown in part 4.6). If you don't use the imaginary part of the impedance, wouldn't a DC measurement be sufficient? Please explain and comment.
4.6. Gas sensing performances of mesitylene
This paragraph title does not have the meaning that the authors intended (it means that mesitylene is a gas sensor). I suggest: Performances (of the sensors) for mesitylene sensing.
The characterisation of sensor performances has weaknesses and a lack of rigour that need to be highlighted and commented on, before publication.
The experimental set-up is rather rudimentary and does not allow a rigorous characterisation of the sensor. Pumping with a vacuum pump removes the mesitylene vapour, but also the oxygen adsorbed on the sensor surface. Thus, when the mesitylene is reintroduced into the chamber, the sensor surface is not necessarily in equilibrium with the oxygen from the air. Thus, the sensor is not under the normal conditions of use of a pollution sensor. In addition, the gas flow in this large volume chamber is not optimised and can lead to artefacts, as shown in the paper below:
ANNANOUCH, Fatima-Ezahra, BOUCHET, Gilles, PERRIER, Pierre, et al. Hydrodynamic evaluation of gas testing chamber: Simulation, experiment. Sensors and Actuators B: Chemical, 2019, vol. 290, p. 598-606.
(A rigorous set up for gas sensors characterisation must be able to inject a regulated flow rate of air, dry or with controlled humidity, to set the baseline, then the gas to be detected, diluted in dry or humid air, at different concentrations.)
Apparently, temperature and humidity are not controlled. So, how did the authors ensure that all the sensors, whose performance they are comparing, were tested under identical temperature and humidity conditions?
Response curves of figures 8,9,10:
- When comparing sensors that may have a different resistance, it is more relevant to use DR/Rair rather than D Please correct.
- As sensor responses are very noisy, it is necessary to evaluate and report the error in the measurement of response and recovery times, as well as the sensor response.
The authors are invited to ad comments about the following points:
- Why are the sensors responses so noisy?
- After exposure to a constant concentration of analyte, the response of a sensor reaches a horizontal plateau. However, in most of the responses shown, the sensor resistance continues to increase with a smaller slope. How do the authors interpret this phenomenon?
- A characterisation of the response of the sensors at different temperatures would increase the value of this paper. Otherwise, a commentary on the possible effect of temperature would be necessary.
- Similarly, a characterisation of the sensors under different relative humidities (10-90% RH) would be necessary, or at least a comment, based on the literature, on the possible effects of humidity.
These 4 points may provide guidelines for future work on these sensors, which could be mentioned in the conclusion.
Author Response
Dear Reviewers
First of all, we would like to thank you for taking your time to review our manuscript, and we do apologise for the typos and grammar mistakes we made during the writing and editing of the manuscript. And I would like to thank you for your comments and suggestions, which have improved our manuscript. We have made changes by considering your comments, which is indicated in track changes in the main text.
Our response to the reviewer's comment is as follows.
Review-1
The characterisation of the materials and the study of the detection mechanism is comprehensive and rigorous, but the characterisation of sensor performances has weaknesses and a lack of rigour.
Firstly, there are a few mistakes to be corrected:
Line 43 : “low signal to noise ratio” I think it's more about high.
Authors respond
We recognised the error, and it is changed on the manuscript. Changed “low” with “high”.
Line 54: “In order to lower the high working temperature.“ This is not a sentence.
Authors respond
Thank you very much and the sentence has been rewritten.
Line 258: “pore size plays a crucial role in the selectivity” I think it's more about sensitivity.
Authors respond
We are aware of our mistake, and it is fixed. The pore sizes play a crucial role in sensitivity and are well supported by the published paper below.
*Mingchun Li et al. Gas sensing properties of cobalt titanate with multiscalepore structure: experimental and simulation. Sensors, 20, pp. 1787, 2020.
Secondly, several issues need to be addressed:
Lines 79-83: Are the physical properties (size distribution, aggregates size…) of the collected CNP reproducible? How did the authors ensure this? Possible influence of ambient humidity?
Authors respond
The CNPs synthesis was conducted in an open-air and the collection was done at a specific height (~1 cm) from the flame and we do always use the same type of candles (same manufacturer) all the time, which allow us to have the same size distribution all the time. Yes, we do understand the effect of environmental factors such as the airflow and the type of candles we use on the size distribution. All the time we do try our best to maintain the same environmental factors.
3.2. Gas sensing setup, sensor fabrication, and testing
Please add a picture of a sensor and specify its dimensions, the width and spacing of the electrodes, the thickness of the sensing layer.
Authors respond
The point is addressed and added to the manuscript.
What is the advantage of using an AC voltage at 25kHz and an RLC bridge to measure only the sensor resistance (as shown in part 4.6). If you don't use the imaginary part of the impedance, wouldn't a DC measurement be sufficient? Please explain and comment.
Authors respond
In this work, we used the AC voltage to avoid polarisation and at 25kHz the carbonaceous based sensors responded well. When we use LCR meter to determine the electrical property of the sensors, we could have been able to change the parameter such as impedance, resistance and capacitances, conductance…. Therefore for this particular experiment, we chose to use resistance and capacitance however we reported the resistance response only.
4.6. Gas sensing performances of mesitylene
This paragraph title does not have the meaning that the authors intended (it means that mesitylene is a gas sensor). I suggest: Performances (of the sensors) for mesitylene sensing.
Authors respond
The suggestion has been addressed.
The characterisation of sensor performances has weaknesses and a lack of rigour that need to be highlighted and commented on, before publication.
The experimental set-up is rather rudimentary and does not allow a rigorous characterisation of the sensor. Pumping with a vacuum pump removes the mesitylene vapour, but also the oxygen adsorbed on the sensor surface. Thus, when the mesitylene is reintroduced into the chamber, the sensor surface is not necessarily in equilibrium with the oxygen from the air. Thus, the sensor is not under the normal conditions of use of a pollution sensor. In addition, the gas flow in this large volume chamber is not optimised and can lead to artefacts, as shown in the paper below:
ANNANOUCH, Fatima-Ezahra, BOUCHET, Gilles, PERRIER, Pierre, et al. Hydrodynamic evaluation of gas testing chamber: Simulation, experiment. Sensors and Actuators B: Chemical, 2019, vol. 290, p. 598-606.
Authors respond
In our experiments, a vacuum pump is used to flash out mesitylene vapour and introduce environmental air into the chamber due to the drop in the pressure that was created by the pump inside the chamber. The vacuum pump sucks the mesitylene vapour from the chamber. This is done at atmospheric pressure, environmental air is allowed to enter at one end and on the other end with the help of a vacuum pump remove the analyte vapour. And then the sensor is allowed to equilibrate with environmental air for 3 min before the next injection.
(A rigorous set up for gas sensors characterisation must be able to inject a regulated flow rate of air, dry or with controlled humidity, to set the baseline, then the gas to be detected, diluted in dry or humid air, at different concentrations.)
Apparently, temperature and humidity are not controlled. So, how did the authors ensure that all the sensors, whose performance they are comparing, were tested under identical temperature and humidity conditions?
Authors respond
All the experiments were done under room temperature and constant relative humidity (42%); however, we included the effect of humidity to see the effect of the sensor performance.
Response curves of figures 8,9,10:
- When comparing sensors that may have a different resistance, it is more relevant to use DR/Rair rather than D Please correct.
Authors respond
We do understand that the response in most cases is defined as ΔRa/Rg however it is equally defined as the relative response as ΔR, in this work we define the response as a relative response, therefore, we choose to use the ΔR.
- As sensor responses are very noisy, it is necessary to evaluate and report the error in the measurement of response and recovery times, as well as the sensor response.
Authors respond
We appreciate your comment and the error bars on sensor response calibration, humidity bar graph and repeatability calibration graph were added.
The authors are invited to ad comments about the following points:
- Why are the sensors responses so noisy?
Authors respond
Yes, our sensor responses (signal) were not a smooth line it is accompanied by fluctuation, there are many reasons for the fluctuations. The most common cause of the fluctuations in the response are well known. It is known that most solid state-based sensors generate noise during measurement, this noise is related to the active layer of the metal oxides and the environment around the sensor. For example, in metal oxide, the level of the noise depends on the stoichiometry of oxygen and the displacement of oxygen atoms and also the electronic charge transport due to the presence of oxygen on the surface of the metal oxide. however, during detection of a gaseous analyte, the response fluctuates due to fluctuation of concentration and distribution of chemical species on the surface of the active material which is due to free carrier’s number and mobility fluctuations. This can be related to the adsorption-desorption (A-D) noise due to the adsorption-desorption of gas molecules. And the second source of noise during measurement is the diffusion noise related to the diffusion of molecules adsorbed on the surface adsorption, and the third source of the shot noise due to the current through the potential barriers at grain boundaries in the sensitive layer. Therefore we do believe all the mentioned sources of noise are responsible for the noisy responses in our results. There is excellent researches were done by:
L.B. Kish et al. / Sensors and Actuators B 71 (2000) 55-59
- Contaret et al IEEE Sensors Journal, vol. 13, no. 3, pp. 980-986
- After exposure to a constant concentration of analyte, the response of a sensor reaches a horizontal plateau. However, in most of the responses shown, the sensor resistance continues to increase with a smaller slope. How do the authors interpret this phenomenon?
Authors respond
The mesitylene has very low vapour pressure ( 2 mmHg (20°C)), usually organic compounds (analyte) with very low vapour pressure the response don’t reach a horizontal plateau, however analyte with high vapour pressure it is easy to get a horizontal plateau. low vapour pressure means a high boiling point (164.7 °C).
- A characterisation of the response of the sensors at different temperatures would increase the value of this paper. Otherwise, a commentary on the possible effect of temperature would be necessary.
Authors respond
The purpose of this research is to reduce the working temperature of the sensors, when sensors run at high temperatures, the active material starts degrading and the life of the sensor reduces. since the active material of our sensors is composed of metal oxide-carbon nanoparticles and polymer, it is believed that at high temperature the active material would decompose.
- Similarly, a characterisation of the sensors under different relative humidities (10-90% RH) would be necessary, or at least a comment, based on the literature, on the possible effects of humidity.
Authors respond
We performed the selected sensors at different humidity levels, 45%, 52% and 61%, and we found there is no significant effect on the sensor performance.

Reviewer 2 Report
This manuscript presents the formation of composites based on NiO-CNPs-CA and their sensing properties on mesitylene. The results provide a potential to use the composites for gas sensors, but the discussion on the observed results and mechanisms is unclearly described. The authors need to address the following comments:
1) 4.1 TEM on Page 5: The authors observed that the synthesized NiO particles show mostly cubic particles probably due to the formation of the cubic phase. NiO particles with spherical and hexagonal structures were also found in TEM results. Does it mean that parts of NiO particles are non-cubic phases?
2) Figure 4: From EDS measurement of CNPs, about 91% carbon and 9% oxygen are detected. The authors need to comment on how oxygen atoms exist, for example, surface adsorption on the carbon surface, doping into carbon bonds, etc.
3) 4.2. Powder X-ray diffraction on Page 7: The authors used the Debye-Scherrer formula to calculate the average NiO particle size, and the size was 21.7 nm. From TEM results (4.1.), the particle size of NiO ranged between 20 and 90 nm, and the distribution graph appears to show the average size of around 50 nm. The authors need to comment on these two values.
4) It would be better to add wave number values inside Figure 6 (a) corresponding to CA.
5) CNPs-CA and NiO-CNPs sensors showed the gas response but did not show linearity. What would be possible reasons?
6) For a composite of three sensing materials (NiO-CNPs-CA), the ratio between CNPs and CA is fixed at 1:3. The authors may add the reason to determine such a ratio. Figure 8 (a) shows only the response of the 3:1 CNPs-CA sensor.
7) In detecting mesitylene from the synthesized composite sensors, it appears that oxygen probably adsorbed on the CNP surface may play a role in resistance change based on FTIR results supporting the formation of CO2. However, the mechanism does not clearly articulate the role of NiO, the reason for the lower sensitivity of Snr 8, possible chemical reactions between oxygen species and mesitylene vapor, etc. For example, types of adsorbed oxygen species are reported to depend on the environment temperature. O2- is a more favorable form at low temperatures and O- is more at higher temperatures. The authors should discuss a possible change of oxygen species on gas sensing mechanisms. Additionally, the cited references (41-43) are related to how oxide sensors respond to gas species. Since the authors claim oxygen on CNPs to be a dominant factor, it is unclear whether the mechanism of oxide surface reaction can be valid to carbon surfaces with a composite structure.
Author Response
Dear Reviewers
First of all, we would like to thank you for taking your time to review our manuscript, and we do apologise for the typos and grammar mistakes we made during the writing and editing of the manuscript. And I would like to thank you for your comments and suggestions, which have improved our manuscript. We have made changes by considering your comments, which is indicated in track changes in the main text.
Our response to the reviewer's comment is as follows.
These 4 points may provide guidelines for future work on these sensors, which could be mentioned in the conclusion.
This manuscript presents the formation of composites based on NiO-CNPs-CA and their sensing properties on mesitylene. The results provide a potential to use the composites for gas sensors, but the discussion on the observed results and mechanisms is unclearly described. The authors need to address the following comments:
1) 4.1 TEM on Page 5: The authors observed that the synthesized NiO particles show mostly cubic particles probably due to the formation of the cubic phase. NiO particles with spherical and hexagonal structures were also found in TEM results. Does it mean that parts of NiO particles are non-cubic phases?
Authors respond
1) We are aware that NiO structures are cubic phases but based on reaction conditions used such as temperature and time can affect the morphology. Both Suresh et. al. and Md Rukidudin et.al., reported mixed morphological structures (spheres and hexagons) , however, their crystallographic nature is a cubic phase. Both published articles used sol-gel synthetic method.
-Suresh Kumar Pandey et.al, Highly facile Ag/NiO nanocomposite synthesized by sol-gel method for mineralization of rhodamine B, Ceramics International, Vol. 46, pp. 8631-8639, 2020.
-Md Rakibuddin et.al, Sol-gel fabrication of NiO and NiO/WO3 based electrochromic device on ITO and flexible substrate, ceramic international, Vol .46, pp. 8631-8639, 2020.
2) Figure 4: From EDS measurement of CNPs, about 91% carbon and 9% oxygen are detected. The authors need to comment on how oxygen atoms exist, for example, surface adsorption on the carbon surface, doping into carbon bonds, etc.
Authors respond
2) Oxygen atoms on CNPs are formed during pyrolytic reactions in exposed atmospheric air. CNPs are the collected smoke produced from the burning of a lighthouse candle, thus the atmospheric oxygen molecules react with a produced hot smoke during collection. Soots are a product of incomplete combustion.
3) 4.2. Powder X-ray diffraction on Page 7: The authors used the Debye-Scherrer formula to calculate the average NiO particle size, and the size was 21.7 nm. From TEM results (4.1.), the particle size of NiO ranged between 20 and 90 nm, and the distribution graph appears to show the average size of around 50 nm. The authors need to comment on these two values.
Authors respond
- The suggestion was taken into consideration, the recalculated particle size from using the Scherrer formula was found to be 46 nm. The distribution graph was drawn using a Gaussian curve and the average particle size was found to be approximately 55 nm. This point was fixed even on the manuscript.
4) It would be better to add wave number values inside Figure 6 (a) corresponding to CA.
Authors respond
4) it is corrected in the main text. The wavenumbers were added inside the Raman spectrum of CA.
5) CNPs-CA and NiO-CNPs sensors showed the gas response but did not show linearity. What would be possible reasons?
Authors respond
The linear relationship between the analyte concentration and the sensor response largely depends on the availability of active sites, if the analyte molecules are larger (in concentration) than the available active site on the sensing materials, then we wouldn’t see a linear relationship, it would be a flat line response. What we see for CNP-CA, at low concentration, the first three injections, the response increases progressively as the concentration increases, there was a linear relationship, however, at higher concentrations of the analyte, that means the last two injections the response didn’t change much from the third injection. This means all the active sites were occupied and it reached a saturation point.
6) For a composite of three sensing materials (NiO-CNPs-CA), the ratio between CNPs and CA is fixed at 1:3. The authors may add the reason to determine such a ratio. Figure 8 (a) shows only the response of the 3:1 CNPs-CA sensor.
Authors respond
6) The combination started from the 3 material composites (NiO, CNPs, CA). Firstly [1:1:1], [1:1:2 ] and [1:1:3] mass ratio of NiO: CNPs: CA were tested for the detection of mesitylene vapour, wherein NiO and CNPs mass were fixed and CA mass was varied. Unfortunately [1:1:1] and [1:1:2] NiO-CNPs-CA mass ratios did not respond to our analyte but while 1:1:3 mass ratio did respond to mesitylene vapour. Then to check the effect of metal oxide within the [1:1:3] NiO-CNPs-CA composite, the CNPs-CA 1:3 mass ratio was fixed and the mass of NiO was varied. And the sensing responses and sensitivities were much improved as compared to binary material systems (CNPs-CA and NiO-CNPs).
7) In detecting mesitylene from the synthesized composite sensors, it appears that oxygen probably adsorbed on the CNP surface may play a role in resistance change based on FTIR results supporting the formation of CO2. However, the mechanism does not clearly articulate the role of NiO, the reason for the lower sensitivity of Snr 8, possible chemical reactions between oxygen species and mesitylene vapor, etc. For example, types of adsorbed oxygen species are reported to depend on the environment temperature. O2- is a more favorable form at low temperatures and O- is more at higher temperatures. The authors should discuss a possible change of oxygen species on gas sensing mechanisms. Additionally, the cited references (41-43) are related to how oxide sensors respond to gas species. Since the authors claim oxygen on CNPs to be a dominant factor, it is unclear whether the mechanism of oxide surface reaction can be valid to carbon surfaces with a composite structure.
Authors respond
Yes, adsorbed oxygen is determining factor in the detection of volatile organic compounds, if the sensing mechanism goes deep oxidation. Although there are different sensing mechanisms, In the case of the polymer, however, the dominant effects in the sensing mechanism are electrostatic, polarization and Lewis acid and base interactions between the analyte molecules and the polymer and/ or adsorption of the analyte molecules, however, is other dominant phenomena in sensing mechanism, by swelling the polymer structure which increases the overall volume and the polymer chains push away from one another, changing the conductivity of the sensors. Therefore, for metal oxides,NiO, however, the adsorption of oxygen on the surface is the most important stapes during the sensing process, those adsorbed oxygen molecular converted to reactive ionosorbed oxygen (see Figure 6 (c) and (d)). Similarly, for CNPs the adsorbed oxygen molecule converts into ionosorbed oxygen. This claim is confirmed with XPS analysis which was added into the main text recently (see Figure 6 (b)).
Reviewer 3 Report
I read your manuscript entitled “Nickel oxide-carbon soot-cellulose acetate nanocomposite for the detection of mesitylene vapour: Investigating the sensing mechanism using an LCR meter coupled to an FTIR spectrometer”. My comments are as follows: 1- English editing is necessary. Some sentences are too long. There are some mistakes in some sentences. 2- Fig.1 & 2 can be combined. 3- XPS is recommended to gain more insight into synthesized materials. 4- In Fig. 5, please put JCPDS cards. 5- y-axis in Fig 7c should be corrected. 6- Please add BJH plots in surface area studies. 7- Table 1 is not suitable in appearance. 8- Please define response and recovery times. 9- Stability, selectivity and repeatability of gas sensors should be evaluated. 10- How about effect of water vapor on the gas response?
Author Response
Dear Reviewers
First of all, we would like to thank you for taking your time to review our manuscript, and we do apologise for the typos and grammar mistakes we made during the writing and editing of the manuscript. And I would like to thank you for your comments and suggestions, which have improved our manuscript. We have made changes by considering your comments, which is indicated in track changes in the main text.
I read your manuscript entitled “Nickel oxide-carbon soot-cellulose acetate nanocomposite for the detection of mesitylene vapour: Investigating the sensing mechanism using an LCR meter coupled to an FTIR spectrometer”. My comments are as follows: 1- English editing is necessary. Some sentences are too long. There are some mistakes in some sentences. 2- Fig.1 & 2 can be combined. 3- XPS is recommended to gain more insight into synthesized materials. 4- In Fig. 5, please put JCPDS cards. 5- y-axis in Fig 7c should be corrected. 6- Please add BJH plots in surface area studies. 7- Table 1 is not suitable in appearance. 8- Please define response and recovery times. 9- Stability, selectivity and repeatability of gas sensors should be evaluated. 10- How about effect of water vapor on the gas response?
- English editing is necessary
Authors respond
We apologise for the typo errors and grammar mistakes we made during the writing and editing of the manuscript. We edited and rewrite some sections of the main text.
- 1 & 2 can be combined.
Authors respond
With respect, Fig. 1 and Fig. 2 are different diagrams with different meanings, thus they cannot be combined, Fig. 1 describes how we perform the sensors testing it is dynamic response however Fig. 2 shows how we perform a static response for the sensing mechanism and sensor where placed in a small cylinder inside the FTIR and exposed to a constant concentration of mesitylene vapour on the sensor.
- XPS is recommended to gain more insight into synthesized materials
Authors respond
Thank you for XPS recommendation and it gave more weight to our manuscript. XPS analysis of the synthesised materials was done, and it was found that carbon soot, nickel oxide and NiO-CNPs-CA composite have reactive oxygen species. The oxidation state of nickel in NiO and NiO-CNPs-CA composite was found to be Ni 2+.
- In Fig. 5, please put JCPDS cards
Authors respond
We agree that it is important to discuss XRD using JCPDS cards and our XRD data was discussed using JCPDS as shown in the discussion paragraphs and clearly described every individual material.
- 5- y-axis in Fig 7c should be corrected
Authors respond
We apologise for our mistake, and it is now corrected.
- Please add BJH plots in surface area studies
Authors respond
We are aware of the BJH plot and it is very important, unfortunately, our BET data is corrupted. We apologise for not putting BJH plot.
- Table 1 is not suitable in appearance
Authors respond
Corrected, the response-recovery times of the sensors are now tabulated using the MDPI nanomaterial template.
- Please define response and recovery times
Authors respond
it is very important to define response and recovery time. We did define the response and recovery time in the main text.
- Stability, selectivity and repeatability of gas sensors should be evaluated.
Authors respond
Stability and repeatability are considered important factors to be studied. Repeatability is studied was added (see Figure 14), and our sensor maintained its stability. We are aware of the selectivity, but our work was more on explaining the sensing mechanism of mesitylene and we used only one analyte, thus the selectivity is not studied (we apologise for that).
- How about effect of water vapor on the gas response?
Authors respond
We performed the selected sensors at different humidity levels, 45%, 52% and 61%, and we found there is no significant effect on the sensor performance.
Round 2
Reviewer 2 Report
The authors have satisfactorily responded to the comments and made the necessary changes to the manuscript.
Reviewer 3 Report
It has been significantly improved and now it can be accepted....